# Effect of Mesh Sensitivity and Cohesive Properties on Simulation of *Typha* Fiber/Epoxy Microbond Test

**Ikramullah [1],\*, Andri Afrizal [1], Syifaul Huzni [1],\* , Sulaiman Thalib [1], H. P. S. Abdul Khalil [2]
and Samsul Rizal [1],\***

[1]  Department of Mechanical Engineering, Universitas Syiah Kuala, Darussalam, Banda Aceh 23111, Indonesia;
    andri85@mhs.unsyiah.ac.id (A.A.); sulaimanthalib@unsyiah.ac.id (S.T.)
[2]  School of Industrial Technology, Universiti Sains Malaysia, Penang 11800, Malaysia; akhalilhps@gmail.com
\*  Correspondence: ikramullah@mhs.unsyiah.ac.id (I.); syifaul@unsyiah.ac.id (S.H.);
    samsul_r@yahoo.com (S.R.)

**Abstract:** The microbond test for natural fibers is difficult to conduct experimentally due to several challenges including controlling the gap distance of the blade, the meniscus shape, and the large data spread. In this study, a finite element simulation was performed to investigate the effects of the bonding characteristics in the interface between the fiber and matrix on the *Typha* fiber/epoxy microbond test. Our aim was to obtain the accurate mesh and cohesive properties via simulation of the *Typha* fiber/epoxy microbond test using the cohesive zone model technique. The axisymmetric model was generated to model the microbond test specimen with a cohesive layer between the fiber and matrix. The cohesive parameter and mesh type were varied to determine the appropriate cohesive properties and mesh type. The fine mesh with 61,016 elements and cohesive properties including stiffness coefficients $K_{nn}$ = 2700 N/mm$^3$, $K_{tt}$ = 2700 N/mm$^3$, and $K_{ss}$ = 2700 N/mm$^3$; fracture energy of 15.15 N/mm; and damage initiation $t_{nn}$ = 270 N/mm$^2$, $t_{tt}$ = 270 N/mm$^2$, and $t_{ss}$ = 270 N/mm$^2$ were the most suitable. The cohesive zone model can describe the debonding process in the simulation of the *Typha* fiber/epoxy microbond test. Therefore, the results of the *Typha* fiber/epoxy microbond simulation can be used in the simulation of *Typha* fiber reinforced composites at the macro-scale.

**Keywords:** mesh sensitivity; cohesive properties; microbond test; *Typha* fiber

## 1. Introduction

Natural fiber-reinforced polymer composites have attracted research interest as replacements for synthetic fibers. Synthetic fibers are not environmentally friendly due to being nonbiodegradable, which causes environmental problems. Therefore, natural fibers have rapidly been developed to reinforce polymer composites. Natural fibers are biodegradable and renewable. Several natural fiber composites have been studied including ramie, kenaf, coir, sisal, hemp, and jute [1–3]. One of the natural fibers that has the potential to reinforce composites is *Typha* fiber. *Typha* is widely available in most countries [4]. *Typha* grows rapidly on wetlands and is often considered as parasite. Although *Typha* is abundantly present in nature, its potential is still undeveloped compared to other natural fibers [5]. *Typha* fiber composites are known to have many advantages, such as good flexibility, light weight, good mechanical strength, low density, and renewability [6,7]. However, several challenges exist to the use of natural fiber-reinforced polymer composite, one of which is the low compatibility between natural fiber and the matrix polymer [8], which causes interlayer gaps and initial cracks along the fiber. The interfacial composite is the contact area between the matrix and the reinforcing fiber that carries the load received by the matrix to the fiber. The structures of the fiber and the matrix

considerably influence the bonding of the fiber–matrix interface, which also affects the characteristics of the composite structure [9–11].

Several testing methods, such as the fiber push-out test, microbond test, and fragmentation test, have been used to evaluate the shear strength of the interface of the composite. The microbond test was developed by Miller et al. [12] and is one of the most widely used test methods for testing fiber–matrix interfacial shear strength (IFSS).

Microbond tests have been commonly used to study the effects of surface treatments on fibers and the interface behavior between matrices and fibers. The microbond test for natural fibers is hard to conduct experimentally due the several challenges such as difficulty in controlling the gap distance of the blade, the meniscus shape, the plastic yield of the matrix, and the reproducibility of sample preparation [13]. Microbond test results are reported cause large variations in test data and the large data spread becomes an obstacle for determining the shear strength of composite interfaces compared to other tests [14]. Therefore, microbond testing with numerical simulation approaches has attracted researchers. Numerical simulation using the finite element method has been used to simplify the investigation of the bonding characteristics on the interface between the fiber and matrix [15,16].

Several reports have been published about microbond test simulations. Ash et al. developed an axisymmetric model to study microbond tests including realistic droplet shapes on the glass fiber–polymer matrix [17]. Kang et al. modeled microbond tests using two-dimensional (2D) axisymmetric models and considered different droplet shapes, such as cylinders and spheres, on the effect of meniscus on carbon–epoxy fibers [14]. Crack propagation during fiber-on-fiber/matrix push-out was modeled using a cohesive zone approach in the volumetric finite element method (FEM) with an axisymmetric model [18]. The cohesive zone model was used to simulate the interphase between fiber and matrix, providing an ideal tool to study the interfacial debonding progress in composites [19]. However, no study has investigated the behavior of interfacial bonds in the microbond test of *Typha* fiber/epoxy. The aim of this study was to determine the accurate mesh and cohesive properties on the simulation of the *Typha* fiber/epoxy microbond test using the cohesive zone model technique.

The *Typha* fiber/epoxy microbond test was simulated using the FEM. The mesh sensitivity and cohesive properties are two parameters that play important roles in microbond test simulation. Therefore, we wanted to determine the appropriate mesh type and cohesive properties to obtain good agreement between simulation and experimental results. We wanted to analyze the mesh sensitivity and cohesive properties of the *Typha* fiber/epoxy microbond test as these have not been reported in the literature.

## 2. Materials and Methods

### 2.1. FEM

The FEM and geometry used were similar to the microbond test experiment, as shown in Figure 1. The commercial finite element software Abaqus 6.14 (Dassault Systèmes Simulia Corp., Johnston, RI, USA) was used to generate the model and analyze the *Typha* fiber/epoxy microbond test. The axisymmetric model was generated to model the microbond test specimen with a cohesive layer between the fiber and matrix with 0.01 mm thickness, as shown in Figure 2, where the embedded length, fiber radius, and droplet height were 1.67, 0.125, and 0.3 mm, respectively. The vise blade was 0.78 mm away from the interface. The vise blade was modelled as a rigid body. The hard contact type was selected as the interaction between the blade and resin. Tie constraints were used to tie fibers to the cohesive layer and matrix. A displacement load of 0.466 mm was applied at the end of the fiber. The magnitude of displacement was obtained from experiments. The edge of the fiber and matrix provided the asymmetrical boundary condition, where only the Y plane axis is free (U1 = U2 = UR3 = 0), while the vise blade is in the fully built-in fixed support condition (U1 = U2 = UR1 = UR2 = UR3 = 0). Finite element analysis was performed on an ASUS computer (AsusTek Computer Inc., Taipei, Taiwan) with an Intel core i9 processor (Intel Corp., Santa Clara, CA, USA) and 16 GB RAM.

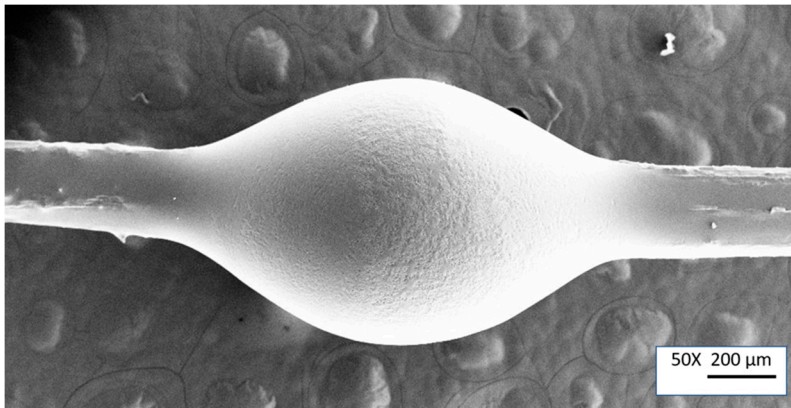

**Figure 1.** The photograph of fiber–matrix interface microbond.

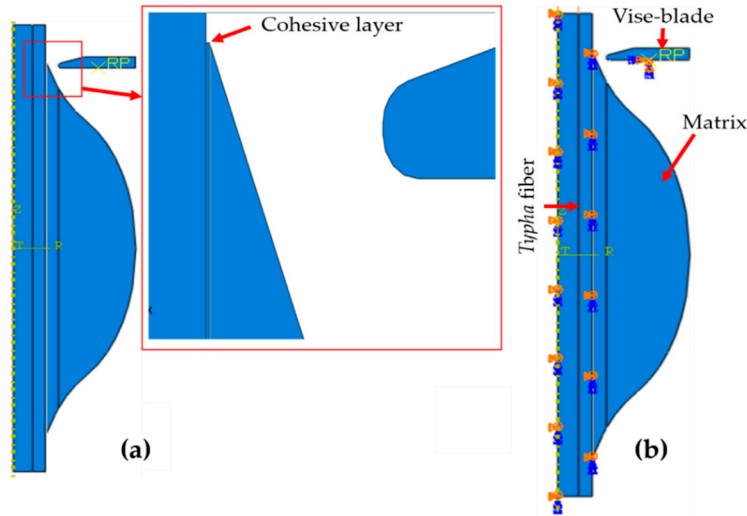

**Figure 2.** (**a**) Microbond test model geometry and (**b**) boundary condition.

The mesh was varied to determine the mesh that produces results near to the experimental microbond test results. The element near the interfacial area was adjusted. The quadrilateral linear CAX4 type and linear triangular CAX3 type were the elements used on the epoxy resin and *Typha* fiber, whereas the elements on the cohesive layer were quadrilateral linear COHAX4 type. The mesh on the *Typha* fiber/epoxy microbond test model is displayed in Figure 3. The numbers of each element in the microbond testing model are presented in Table 1.

**Table 1.** The number of elements used in the model.

| Mesh Type | Seed Mesh | *Typha* Fiber and Matrix | | Cohesive Layer | Total |
| --- | --- | --- | --- | --- | --- |
| | | CAX4 | CAX3 | COHAX4 | |
| Coarse | 0.005 | 2256 | 183 | 1675 | 4114 |
| Medium | 0.003 | 5896 | 230 | 1675 | 7801 |
| Fine | 0.001 | 58,324 | 1017 | 1675 | 61,016 |
| Very Fine | 0.0005 | 18,8491 | 3145 | 1675 | 193,311 |

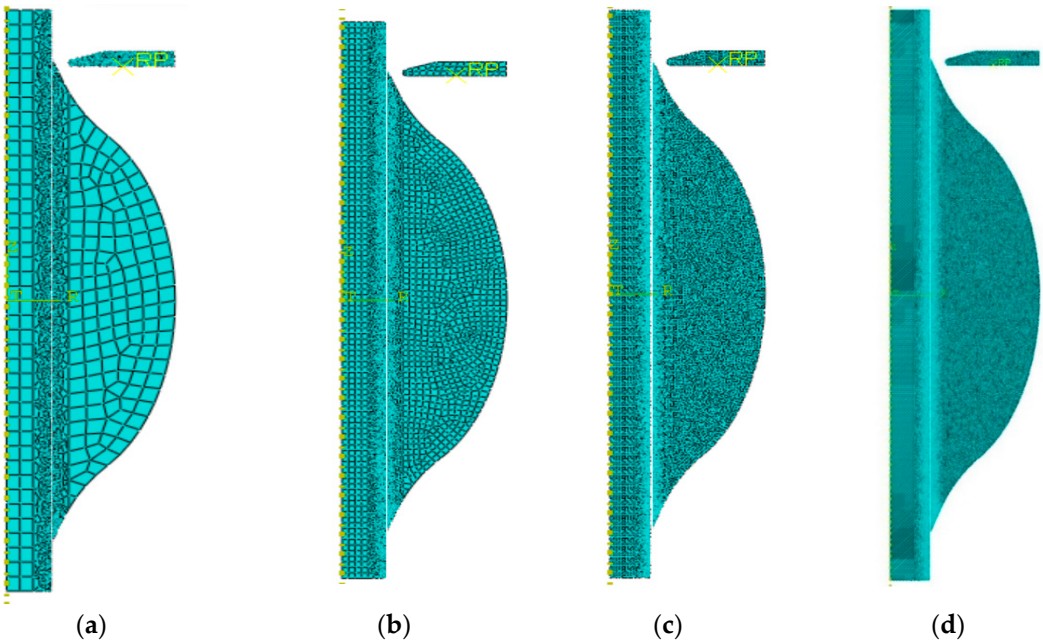

| (a) | (b) | (c) | (d) |

**Figure 3.** (**a**) Coarse, (**b**) medium, (**c**) fine, and (**d**) very fine mesh.

## 2.2. Material Properties

Dugdale [20] and Barenblatt [21] introduced the cohesive zone model in 1960 and 1962, respectively, when solving the problem of cracking of perfect brittle material. Three modes of fractures, opening mode, sliding mode, and tear mode, can appear at the interface between the fiber and matrix when the loading conditions are applied to the composite. The cohesive zone model specifies three bond conditions for each fracture mode: elastic bond, debonding, and debonded conditions. All the fracture modes can be characterized by a traction–separation (*t*–*δ*) curve as shown in Figure 4. The condition of the elastic bond is determined by the interfacial stiffness (*K*), which is the elastic constitutive matrix that relates *t* and *δ*, as shown in Equation (1).

$$\mathbf{t} = \mathbf{K}\delta$$
$$\left\{ \begin{array}{c} t_n \\ t_s \\ t_t \end{array} \right\} = \left[ \begin{array}{ccc} K_{nn} & K_{ns} & K_{nt} \\ K_{ns} & K_{ss} & K_{st} \\ K_{nt} & K_{st} & K_{tt} \end{array} \right] \left\{ \begin{array}{c} \delta_n \\ \delta_s \\ \delta_t \end{array} \right\} \tag{1}$$

where *n* is normal mode, *s* is shear mode, and *t* is tear mode.

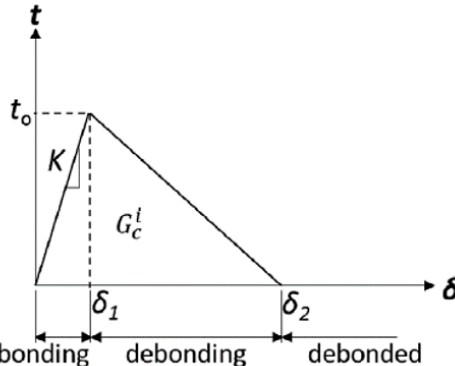

**Figure 4.** Traction–separation curve. $K$ is interfacial stiffness, $G_c^i$ is interfacial fracture energy, $t_o$ is interfacial strength, and $\delta$ is the separation.

The material properties are listed in Table 2. *Typha* fiber properties were obtained from our previous research [22] and the epoxy resin properties were obtained from Selmi [23].

**Table 2.** Material properties *Typha* fiber and epoxy resin.

| Material | Young's Modulus (GPa) | Poisson's Ratio |
|---|---|---|
| *Typha* fiber | 0.88 | 0.35 |
| Epoxy resin | 3.7 | 0.4 |

Adjusting the stiffness parameters and damage criteria for cohesive elements is convenient with finite element software [15]. Therefore, the cohesive properties were varied to obtain agreement of the simulation results with the experimental *Typha* fiber/epoxy microbond results. Table 3 lists the various cohesive properties. These values, obtained based on several simulations, were accepted as reported using the same method of Dadej [24] and Potukuchi [25].

**Table 3.** Variations in cohesive properties.

| Variations | Stiffness Coefficients (N/mm$^3$) | | | Fracture Energy (N/mm) | Damage Initiation (N/mm$^2$) | | |
|---|---|---|---|---|---|---|---|
| | $K_{nn}$ | $K_{ss}$ | $K_{tt}$ | $G_{Ic}$ | $t_{nn}$ | $t_{ss}$ | $t_{tt}$ |
| VR1 | 2700 | 2700 | 2700 | 14.65 | 270 | 270 | 270 |
| VR2 | 2700 | 2700 | 2700 | 15.15 | 270 | 270 | 270 |
| VR3 | 3000 | 3000 | 3000 | 15.00 | 250 | 250 | 250 |
| VR4 | 3500 | 3500 | 3500 | 14.00 | 270 | 270 | 270 |
| VR5 | 5000 | 5000 | 5000 | 10.50 | 267 | 267 | 267 |
| VR6 | 8000 | 8000 | 8000 | 5.50 | 200 | 200 | 200 |
| VR7 | 10,000 | 10,000 | 10,000 | 5.00 | 250 | 250 | 250 |

## 3. Mesh Sensitivity Analysis

The microbond test has several parameters that must be considered, including fiber diameter, embedded length, meniscus angle, contact angle, and the distance between the vise blade and the specimen. Cohesive parameters must be determined first to produce suitable cohesive properties. When accomplished, the experimental results would be similar to the experimental results [15]. The mesh has an important role in finite element analysis for determining the accuracy of the simulation results to ensure good agreement with the experimental results.

The force–displacement curve was obtained from the mesh sensitivity study in the microbond test model, as shown in Figure 5. A, B, C, and D represent the coarse, medium, fine, and very fine mesh, respectively, corresponding to P, Q, R, and S, respectively, which represent computational time. We found that the higher the number of the elements, the lower the reaction force; this indicated that mesh selection affected the simulation result. Figure 5 shows that the coarse mesh had the highest force, which was 9.7 N; the lowest force on the very fine mesh was 1.5 N. The maximum forces for the medium and fine meshes were 7.5 and 2.9 N, respectively.

The results of testing several cohesive properties on the coarse, medium, fine, and very fine mesh are shown in Figure 6a–d, respectively. The force–displacement curves for the cohesive properties were compared with the experimental results. The experimental curve decreased dramatically after reaching the maximum force, with a maximum force value of 2.6 N. The maximum force in the microbond simulation test with fine mesh had a maximum force resembling the experimental results, as shown in Figure 5.

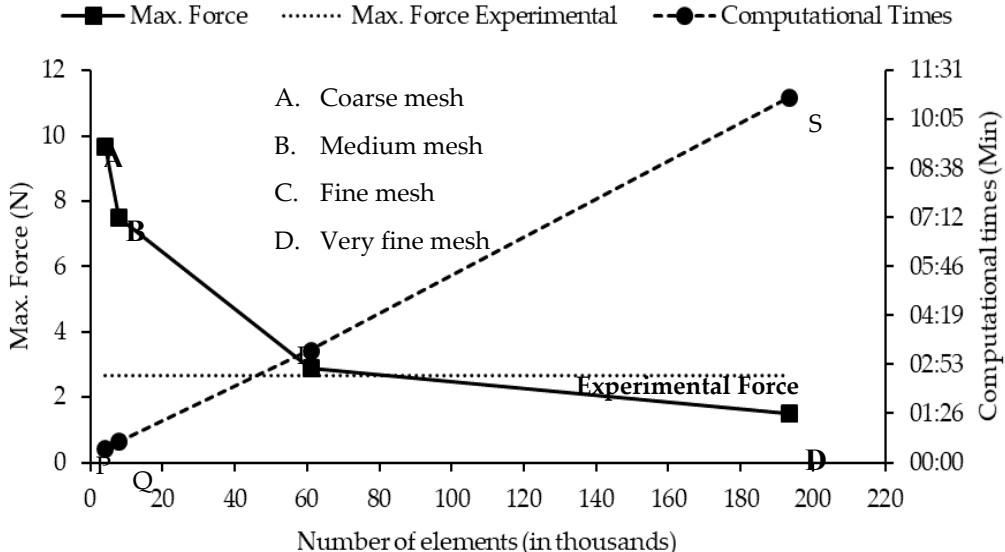

**Figure 5.** Maximum forces and computational times vs. number of elements.

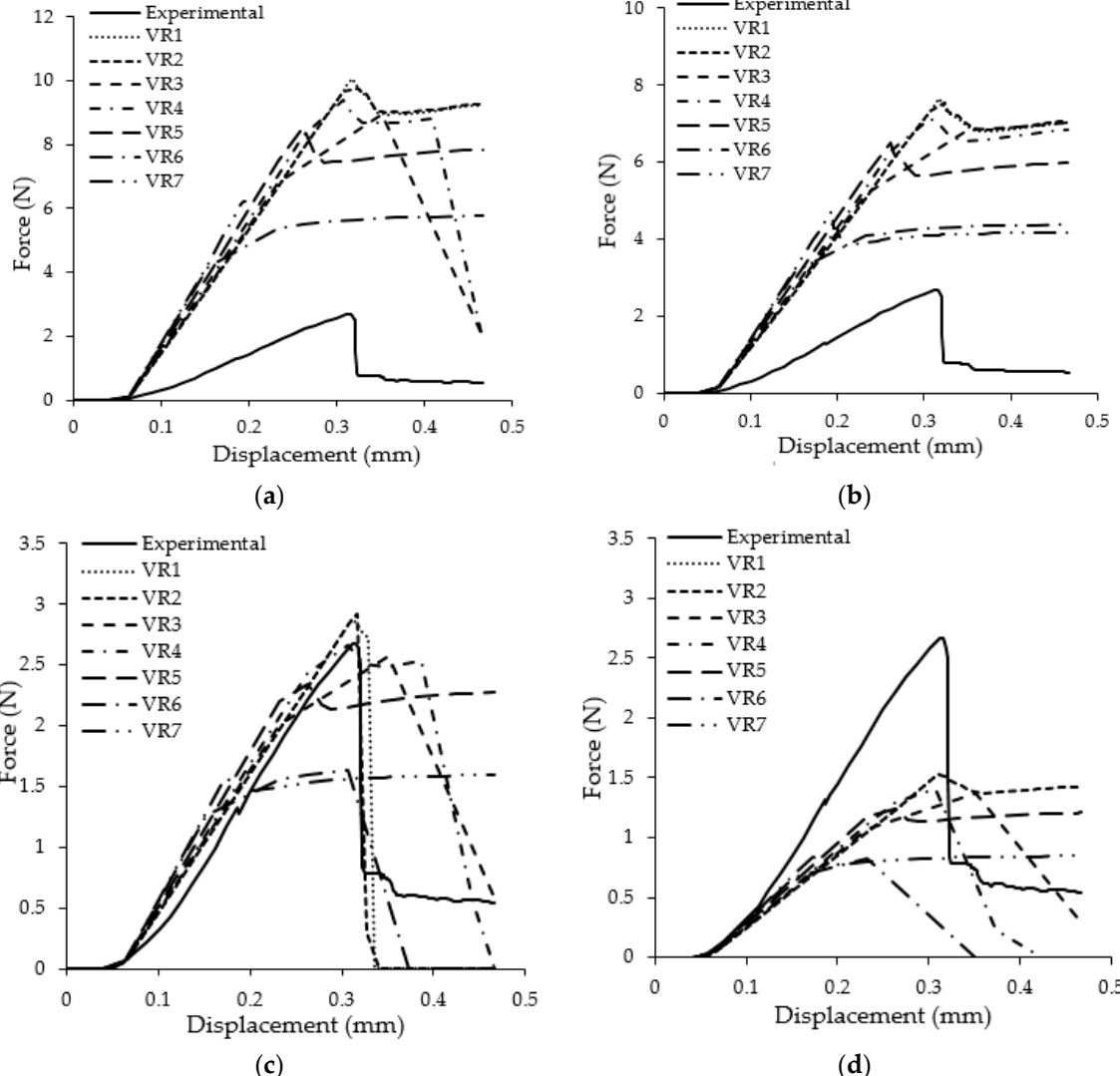

**Figure 6.** The load displacement curve of properties cohesive variations on (**a**) coarse, (**b**) medium, (**c**) fine, and (**d**) very fine mesh compared with experimental values.

In the simulation, the damage initiation parameter value was varied from 200 to 270 N/mm². The variation of the cohesive properties with coarse mesh showed that the maximum force was around 5–10 N. This maximum force exceeded the value obtained from the experiment. For medium mesh, the maximum force was 4–7 N. Based on the image, VR3, VR4, and VR6 show that the curve decreased after reaching maximum force. The simulation of the *Typha* fiber/epoxy microbond test with various cohesive properties on the fine mesh is shown in Figure 6c. We found that the curve was similar to the experimental curve; almost all curves had a maximum force close to the experimental value except for VR6 and VR7. The VR2 curve was the most similar to the experimental results. Figure 6d depicts the results of various cohesive properties on very fine mesh, showing that the maximum force is below the experimental result. Only the VR4 curve decreased after reaching the maximum force.

Based on finite element theory, element size is a critical aspect related to the result accuracy. The smaller the element size, the more accurate the analysis results. However, in certain cases, a smaller element size produces a higher calculation value compared to actual value obtained from the experiments [26]. In this situation, a limit must be set on the percentage error of the results obtained from calculations to determine the appropriate element size [27]. In our study, the least error occurred with the fine mesh.

The incorrect selection of cohesive properties in the microbond test simulation leads to results incompatible with the experimental values. Dadej and Sadowski analyzed the response of the cohesive zone model parameters on glass/epoxy composite and found that incorrectly choosing the damage initiation parameter causes an unstable growth of crack length [24]. Schellekens and de Borst [28] reported that excessive stiffness coefficient parameter values produce unstable analysis, which affects the numerical computation time. The selection of a suitable mesh is also important. As reported by Soto [29], incorrect selection of the number of elements generates incorrect crack initiation and propagation, causing poor agreement of simulation results with experimental results.

From this result, the interfacial shear strength (*IFSS*) of *Typha* fiber/epoxy was obtained using Equation (2) [30].

$$IFSS = \frac{F_{max}}{\pi d_f L_{em}} \tag{2}$$

where $F_{max}$ is the maximum force when debonding occurs between the fiber and matrix, $d_f$ is fiber diameter, and $L_{em}$ is embedded length.

The interfacial shear strength obtained from the simulation results of microbond testing with fine mesh was 4.4 MPa; the interfacial shear strength from the experimental results is 3.2 MPa per previous research [22]. The sensitivity mesh study revealed that fine mesh with 61,016 elements is the most suitable for *Typha* fiber/epoxy microbond test simulation. Therefore, we concluded that the cohesive properties and the type of mesh used in this simulation produced results close to the experimental results.

## 4. Interfacial Debonding

Figure 7 displays the microbond test simulation results, which show the force and displacement of *Typha* fiber/epoxy. The force–displacement curve illustrates that the matrix is still attached to the fiber until a maximum force of 2.9 N at 0.31 mm displacement. After reaching the maximum force, the line on the curve decreases sharply, which reveals debonding has occurred. As illustrated in Figure 8, the debonding phenomenon is similar to the results of the simulation conducted by Hao et al. [31] for a carbon fiber/bismaleimide composites microbond test, but with a different magnitude.

Figure 8 depicts a contour plot of shear stress in a simulation of *Typha* fiber/epoxy composite. When the vise blade hit the matrix first, we observed that the stress at the resin tip is concentrated at the vise blade tip. The shear stress contour spreads from the epoxy resin to the cohesive layer, which was modeled as the interfacial region between the fiber and matrix. Then, the stress was distributed to the fiber, as shown in Figure 8a. In the next step, more shear stress appeared on the fiber and continued to propagate down through the interface area, as shown in Figure 8b. The moment immediately before

debonding, depicted in Figure 8c, illustrates the last step, which is the shear stress concentration moving into the bottom of the test specimen before full debonding occurred.

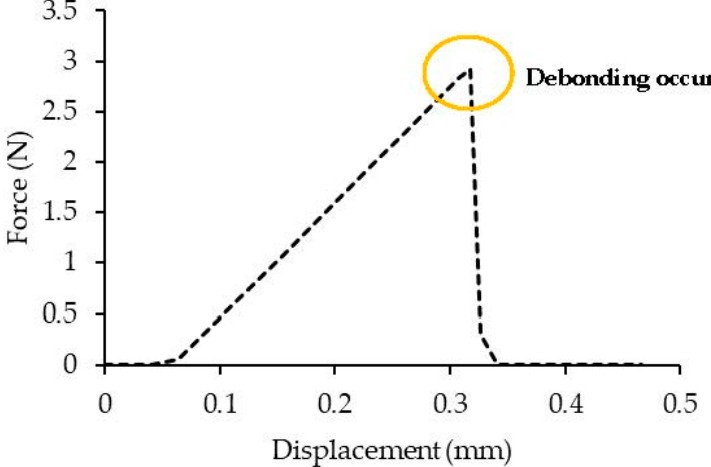

**Figure 7.** Debonding process.

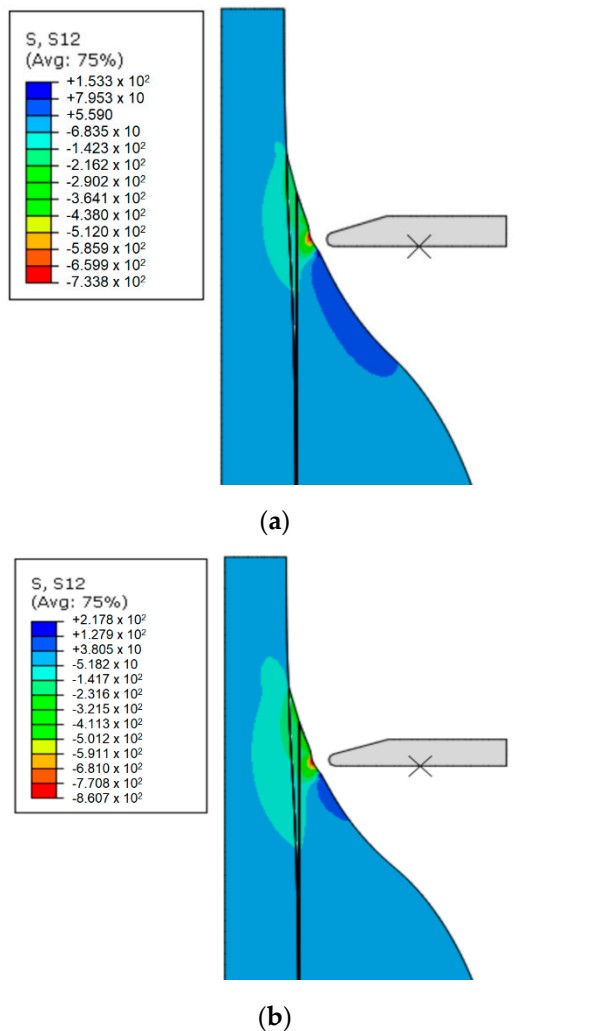

**Figure 8.** *Cont.*

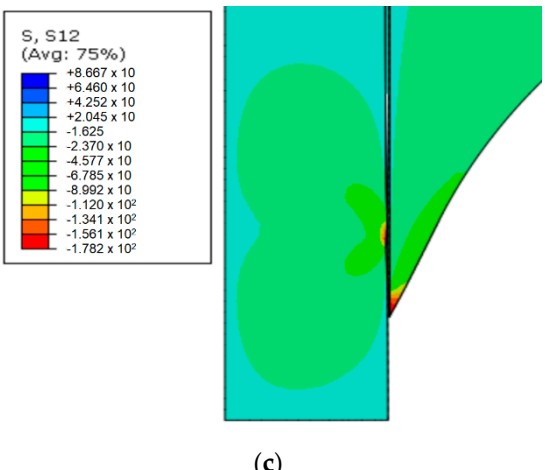

(**c**)

**Figure 8.** Contour plot of shear stress (MPa): (**a**) vise blade first hit matrix, (**b**) then hit the matrix, and (**c**) immediately before debonding.

Figure 9 illustrates the maximum traction damage initiation criterion (MAXSCRT), which is a scalar parameter where total failure is equal to one. The blue and red contours indicate lower and higher MAXSCRT values, respectively. At a step time value of 0.5, red and blue contours were valued at 0.697 and $3 \times 10^{-5}$, respectively, which means failure at the upper cohesive layer has not started. When the step time was 0.689, the upper cohesive layer started to fail due to the value of the red contour reaching 1, while the blue contour was $5 \times 10^{-5}$. The cohesive layer totally failed after a step time of 0.7, as indicated by both the value of red and blue contours reaching one.

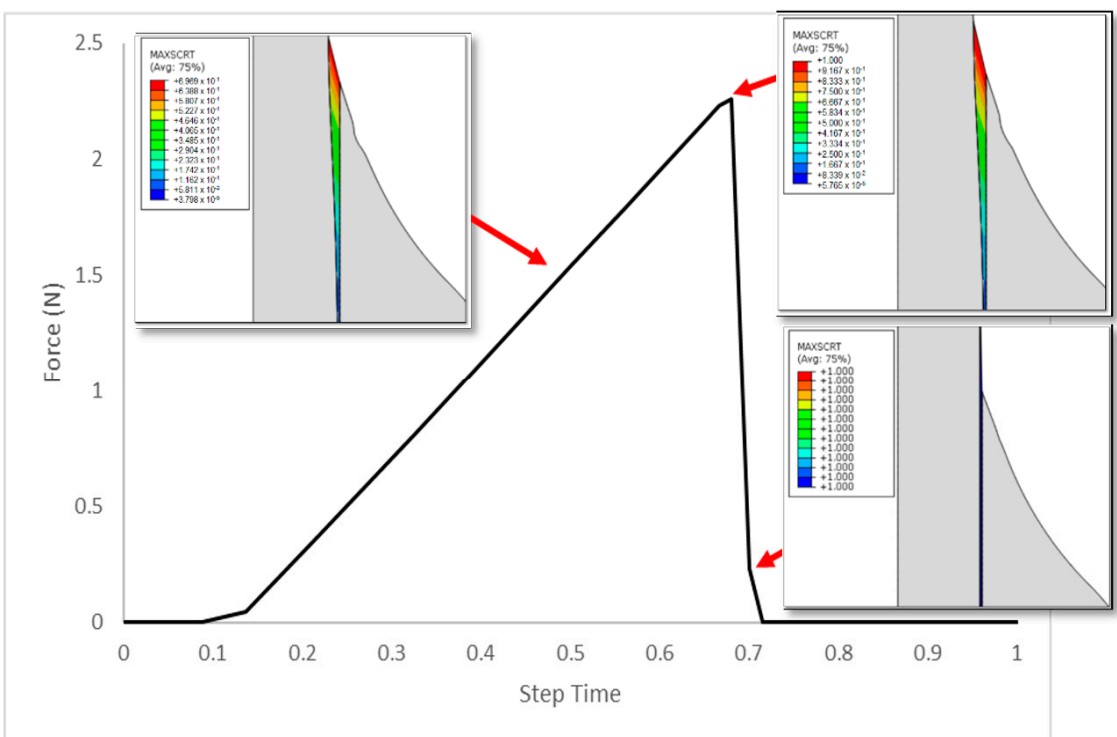

**Figure 9.** Maximum traction damage initiation criterion.

We found that the appropriate mesh type and cohesive properties for the simulation of the *Typha* fiber/epoxy microbond test. The debonding process of the interfacial area between the fiber and the matrix can be clearly described by the proposed model. The mesh type and cohesive properties that

matched the experimental results can facilitate further work on understanding the interfacial behavior of *Typha* fiber and epoxy at the macro scale.

## 5. Conclusions

In this study, we conducted *Typha* fiber/epoxy microbond testing simulations with several different cohesive properties and mesh types. From the results, we concluded:

1.  Fine mesh with 61,016 elements is the most suitable type for the *Typha* fiber/epoxy microbond simulation test based on the comparison of simulation and experimental results.
2.  The simulation results agreed well with experimental results of load and displacement curves and interfacial shear strength values when cohesive properties including stiffness coefficients ($K_{nn}$ of 2700 N/mm$^3$, $K_{tt}$ of 2700 N/mm$^3$, and $K_{ss}$ of 2700 N/mm$^3$), fracture energy of 15.15 N/mm, and damage initiation ($t_{nn}$ = 270 N/mm$^2$, $t_{tt}$ = 270 N/mm$^2$, $t_{ss}$ = 270 N/mm$^2$), with fine mesh.
3.  The cohesive zone model can describe the debonding process during the simulation of the *Typha* fiber/epoxy microbond test.

From this research, we expect that future *Typha* fiber simulation processes on a macro-scale will be able to directly use the mesh parameters and cohesive properties reported here. In future work, the effect of the blade gap distance on the *Typha* fiber/epoxy microbond test will be tested.

**Author Contributions:** Conceptualization, S.R., and H.P.S.A.K.; methodology, I.; software, A.A.; validation, S.H.; formal analysis, A.A.; investigation, I.; resources, S.T.; data curation, S.H.; writing—original draft preparation, I.; writing—review and editing, S.H. and H.P.S.A.K.; visualization, A.A.; supervision, H.P.S.A.K.; project administration, S.R. and H.P.S.A.K.; funding acquisition, S.R. All authors have read and agreed to the published version of the manuscript.

**Funding:** This work was supported by Ministry Research, Technology and Higher Education of Republic of Indonesia by Program Magister Menuju Doktor untuk Sarjana Unggul (PMDSU) scheme No. 72/UN11.2/PP /SP3/2018.

**Acknowledgments:** The authors gratefully thankful to the World Class Professor (WCP) Program 2018 and 2019.

**Conflicts of Interest:** The authors declare no conflict of interest.

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
