# Peer review of "Effect of Mesh Sensitivity and Cohesive Properties on Simulation of Typha Fiber/Epoxy Microbond Test"

_computation, doi:10.3390/computation8010002_

Round 1
Reviewer 1 Report
This manuscript presents finite element (FE) simulations of the microbond test of typha fibre and epoxy resin. The manuscript reports the methodology to create the FE models using cohesive elements between the fibre and the matrix for the debonding, a mesh sensitivity analysis, calibration of model parameters and validation with experimental results. The manuscript is concise with results about the debonding process of the typha fibre embedded in epoxy resin. In general, the results are clear. The manuscript needs improvement and the following comments should be addressed before it can be recommended for publication:
Abstract
-The first phrase of the abstract is misleading since the FE model does not take into account viscous relaxation. This phrase should be rewritten.
General comment
-The manuscript has several grammar and syntax errors and typos. The manuscript should be checked thoroughly for grammatical mistakes and proofread by a native English speaker before it can be recommended for publication.
Introduction
-Figure 3 and Equation 2 and the text related to them should be moved to the Section 2.2.
-Equation 1 should be moved to the end of Section 3, where the interfacial shear strength is presented.
-The authors should add a paragraph at the end of the introduction where they emphasize what is new and unique is this work that has not been done before. i.e. what is the novelty?
Section 2
Section 2.1
-It would be useful if the authors add a photograph with scale of the microbond test similar to Figure 1 in their previous publication (Tailoring the Effective Properties of Typha Fiber Reinforced Polymer Composite via Alkali Treatment).
Section 2.2
-The authors should provide a justification of the range of values used for the properties of the cohesive elements in Table 3. Please provide some references for this.
Section 3
-Figure 4 should be improved. Axis lines should be black colour. Please remove the outer black frame.
-Figure 5 should be improved. Axis lines should be black colour. Legend and axis font size should be bigger. Please remove the outer black frames.
-The authors show in Fig. 5c that the numerical curves are similar to the experimental curve when the fine mesh and variation VR2 are used; however, the data in Fig. 4 suggest that mesh convergence is achieved with the Very Fine Mesh. The authors should provide explanations for this observation and they should justify with some references the use of the Fine Mesh instead of the Very Fine Mesh.
Section 4
-Figure 6 should be improved. Axis lines should be black colour. Axis font size should be bigger. Please remove the outer black frame.
-The authors mention “which is similar to the results of the simulation carried out on carbon fiber/bismaleimide composites microbond test.”. The authors should specify what is similar and how is similar since the referenced microbond test is performed for a different fibre/matrix system.
-At the end of Section 4, the authors should add a discussion paragraph emphasizing the main results and discuss the advantages and disadvantages of the propose model.
Conclusions
The conclusions should be extended emphasizing the main results and discuss briefly the advantages and disadvantages of the propose model and future work.
Reviewer 2 Report
My comments are as follows.
Line 54: It should be Lem instead of Lmax. Equation 2: Please add a description for subscript t, n, and s. Figure 1: Please add a description for parameters indicated on the figure. Line 96: Is there any reason for defining displacement of 0.466mm as loading condition? Line 97: The conditions is specific for ABAQUS. Can you explain it more generally? In general, finer mesh provides better result than coarse one. However, in this study very fine mesh shows worse result than those using fine mesh. Other factor seemed to be more dominant in very fine mesh. Authors should give a comment on it. In Figure 4, the difference of very fine mesh to fine mesh is about 30% smaller. However, in Figure 5, the difference becomes larger of about 50% smaller. Did you know the cause of this phenomenon? The experimental results (Figure 5) should vary among specimens. Can you show us the variation of the experimental results? Figure 7: What is the unit of the shear stress? Please provide the details of the computational environment used in this study, including the computational time for each condition.Author Response
Please see the attachment.

Round 2
Reviewer 1 Report
The authors have addressed most of the reviewer's comments; however, some of the suggestions have not been properly addressed. The paper cannot be recommended for publication until the following comments have been properly addressed:
The first phrase of the abstract is misleading since the FE model does not take into account viscous relaxation. This phrase should be rewritten.
(The authors are still using the word viscous relaxation in the abstract; however, the FE model does not take into account this. This word should be avoided).
The manuscript has several grammar and syntax errors and typos. The manuscript should be checked thoroughly for grammatical mistakes and proofread by a native English speaker before it can be recommended for publication.
(The whole paper still requires "Extensive editing of English language and style required". There are many sentences throughout the manuscript that need to be rewritten, for example in the abstract: "The microbond test specimen was generated an axisymmetric model" or in the introduction "Synthetic fibers are not environmentally friendly due to nonbiodegradable, it will cause environmental problems.". Unless the English language is adequately improved, the paper cannot be accepted for publication).
The authors show in Fig. 5c that the numerical curves are similar to the experimental curve when the fine mesh and variation VR2 are used; however, the data in Fig. 4 suggest that mesh convergence is achieved with the Very Fine Mesh. The authors should provide explanations for this observation and they should justify with some references the use of the Fine Mesh instead of the Very Fine Mesh.
(The authors response to this comment was “The solution was differed from the convergence theory due to the nonlinearities of the case. In fact, the finer mesh the larger stiffness matrix [26]. Refs [27] and [28] prefer to use fine mesh instead of very fine mesh.”. This response is not satisfactory since Refs [27] and [28] used Fine/Medium Mesh instead of Very Fine mesh because their results converged at those mesh sizes so there was no need for a finer mesh. However, in this work the chosed mesh is not the mesh that provides convergence according to Figure 5. The results are clearly mesh dependent. The authors should properly address this comment with some adequate references to support their explanations. They should also explain why the results are mesh dependent and how their predictions can be taken as useful despite this fact.
Reviewer 2 Report
Thank you for revising the manuscript.
Author Response
Thank you for giving your valuable comments and accepting this paper
Round 3
Reviewer 1 Report
The authors have satisfactorily addressed the reviewer's comments. I recommend the manuscript to be published as is.